# Associations between defensive victim-blaming responses (DARVO), rape myth acceptance, and sexual harassment

**Sarah J. Harsey**[1,2]*, **Alexis A. Adams-Clark**[2,3,4], **Jennifer J. Freyd**[2,3]

**1** School of Psychological Science, College of Liberal Arts, Oregon State University–Cascades, Bend, Oregon, United States of America, **2** Center for Institutional Courage, Seattle, Washington, United States of America, **3** Department of Psychology, College of Arts and Sciences, University of Oregon, Eugene, Oregon, United States of America, **4** Department of Psychiatry and Human Behavior, The Warren Alpert Medical School, Brown University, Providence, Rhode Island, United States of America

* sarah.harsey@osucascades.edu

## Abstract

DARVO (Deny, Attack, Reverse Victim and Offender) is a response frequently exhibited by perpetrators of wrongdoing after being confronted or held accountable for their harmful behaviors. Consistent with the original conceptualization of DARVO as a strategy used by sex offenders to deflect blame and responsibility, sexual violence survivors report experiencing DARVO from their perpetrators following an assault. The purpose of the current study was to extend research on the connections between DARVO and sexual violence. We examined whether people who use DARVO as a means of responding to confrontations involving a range of wrongdoings also engage in behaviors and ascribe to beliefs that contribute to sexual violence. A sample of 602 university students was recruited to test hypotheses predicting positive associations between individuals' use of DARVO responses, sexual harassment perpetration, and acceptance of rape myths. Supporting predictions, small but positive correlations emerged between study variables. Data from a second sample of 335 community adults from MTurk were analyzed to replicate findings from the undergraduate sample. Results from the community sample also revealed significant associations between DARVO use, sexual harassment perpetration, and rape myth acceptance. Findings offer further confirmation of a link between DARVO and sexual violence and suggest this defensive response is part of a larger worldview that justifies participation in sexual violence and blames victims.

## Introduction

Sexual violence is a prevalent and significant social issue. More than half of women and approximately 31% of men living in the US report experiencing at least one form of sexual violence at some point during their lives [1]. People with marginalized identities, such as non-heterosexual individuals [2, 3], individuals with non-binary gender identities [4], and those belonging to minoritized racial or ethnic groups [1, 5], face a greater risk of sexual

**Data Availability Statement:** The data underlying the results presented in the study are available from Open Science Framework: https://osf.io/

fny7z/?view_only=
128890beedda4bc4bc88c9463cafe067.

**Funding:** This research project was funded by the Center for Institutional Courage (https://www.institutionalcourage.org/), which disbursed and used research funds granted to J.J.F. by the Clayman Institute for Gender Research at Stanford University (https://gender.stanford.edu/). S.J.H. received a postdoctoral fellowship stipend from the Center for Institutional Courage. The funders had no role in study design, data collection and analysis, decision to publish, or preparation of the manuscript.

**Competing interests:** I have read the journal's policy and the authors of this manuscript have the following competing interests: Jennifer J. Freyd, PhD, is the Founder and President of the Center for Institutional Courage and Professor Emerit of Psychology at the University of Oregon. She is also a Member of the Advisory Committee, 2019-2023, for the Action Collaborative on Preventing Sexual Harassment in Higher Education, National Academies of Science, Engineering, and Medicine. Freyd is the author of the Harvard Press book Betrayal Trauma: The Logic of Forgetting Childhood Abuse. Her most recent book Blind to Betrayal, co-authored with Pamela J. Birrell, was published by John Wiley. For these books Freyd receives royalties. She is often paid honoraria for presentations and she has served as an expert witness on some legal cases for both profit and nonprofit law firms and she has consulted for some governmental and nonprofit organizations. This does not alter our adherence to PLOS ONE policies on sharing data and materials.

victimization. The costs associated with experiencing sexual violence, which includes a range of unwanted sexual behaviors such as sexual assault and sexual harassment, are steep. For instance, nearly three-quarters of victims meet the diagnostic criteria for PTSD in the month following a sexual assault [6]; elevated rates of anxiety disorders, depressive disorders, substance use disorders, and eating disorders [7] are similarly associated with sexual victimization. Moreover, somatic issues (e.g., chronic pain, sleep disturbances) and general health are typically worse among people who have experienced sexual violence [1, 8, 9].

Sexual harassment is a prevalent type of sexual violence that is experienced across a variety of contexts. Although definitions vary [10], sexual harassment typically includes unwanted sexual advances (verbal or physical), requests for sexual favors, and gender harassment (e.g., derogatory or offensive remarks about gender) [11]. In the US, one in three women and one in nine men report experiencing sexual harassment in a public space [1]. Within institutions of higher education, this rate is higher: approximately 75% of transgender, genderqueer, or gender nonconforming undergraduates, 62% of cisgender female undergraduates, and 43% of cisgender male undergraduates enrolled in colleges and universities have been targets of sexual harassment [12]. As with other forms of sexual violence, sexual harassment is associated with a number of adverse consequences, including psychological distress, depression, and alcohol misuse [13].

The current study seeks to quantitatively explore an understudied factor that may aid in the perpetration of sexual violence: DARVO (Deny, Attack, Reverse Victim and Offender). DARVO is a defensive behavior characterized by victim blaming and deflecting responsibility for wrongdoing. By intentionally distorting narratives of sexual violence in their favor, perpetrators may increase their chances of avoiding consequences for committing sexually violent acts. We therefore aimed to test for an association between individuals' use of DARVO and their engagement in sexual harassment, a common form of sexual violence. Moreover, as myths about rape reflect victim-blaming attitudes, we also examined DARVO use and acceptance of rape myths. We predicted that DARVO use would be positively and significantly associated with sexual harassment perpetration and rape myth acceptance in community and undergraduate samples.

## DARVO

DARVO (Deny, Attack, Reverse Victim and Offender) is a defensive response style exhibited by perpetrators of sexual violence when confronted or held accountable for their abusive actions [14]. When using DARVO, perpetrators deny committing any wrongdoing (or deny that their actions have caused harm), attack their victims' credibility, and position themselves as the "real" victims of lies or false accusations. DARVO represents perpetrators' attempts to deflect blame and responsibility by casting victims as untrustworthy and often ill-intentioned narrators. DARVO has two targets: 1) the victim, and 2) third-party observers. DARVO seeks to instill confusion and promote silence (i.e., non-disclosure) among victims; among observers, DARVO functions as a means for perpetrators to recruit sympathetic supporters who will believe the perpetrator and condemn the victim. DARVO occurs primarily as an interpersonal response [14], although perpetrators sometimes engage in litigation against their victims, such as defamation lawsuits [15], in an escalation of this tactic.

Research finds that DARVO is a common response to being confronted or held accountable for wrongdoing. A study examining the prevalence of DARVO asked 138 undergraduates to share their experiences confronting someone else over a wrongdoing [16]. Nearly 72% of the sample reported the person they confronted had used phrases representing all three elements of DARVO–denials, attacks, and reversals–during the confrontation. The wrongdoings

reported by the sample encompassed a variety of interpersonal mistreatment, which ranged from social transgressions (e.g., having a secret betrayed by a family member or close friend) to more serious cases of abuse (e.g., sexual assault and child abuse), suggesting DARVO is used in contexts both in and outside of sexual violence [16]. Further research examining the prevalence of DARVO suggests that victims of sexual violence are also likely to be exposed to this tactic. In a study of 89 college women who experienced campus sexual assault, approximately half reported their perpetrators used elements of DARVO in conversations following the assault [17]. In both studies, experiencing DARVO was found to be associated with self-blame–the more DARVO victims experience from their perpetrators, the more self-blame they feel [16, 17]. For victims of sexual violence, self-blame is related to outcomes that hinder recovery, such as non-disclosure [18] and an increased risk of developing PTSD [19]. This underscores the concerning relationship between victims' DARVO exposure and feelings of self-blame.

Beyond targeting victims, DARVO can also effectively influence observers' perceptions of victims and perpetrators. In an experiment, researchers used a series of fictional vignettes to evaluate how DARVO impacts observers' ratings of victim and perpetrator believability, abusiveness, and responsibility in a sexual assault scenario [20]. Participants– 230 undergraduates–were randomly assigned to view a fictional perpetrator vignette recounting the assault that either contained DARVO responses or one that did not. In the DARVO version of the vignette, the perpetrator denied committing the assault, attacked the credibility of the victim by alleging the victim fabricated the assault as a form of retaliation, and made claims of reputational harm stemming from a false accusation. Victim vignettes were identical between conditions and recounted the assault through the victim's perspective. Compared to those who were not exposed to perpetrator DARVO, participants in the DARVO condition rated the perpetrator as more believable, less responsible, and less abusive; DARVO-exposed participants also rated the victim as less believable and more abusive. Participants were also asked to indicate whether they believed the perpetrator and the victim should face punishment for their actions. Individuals in the DARVO condition, in comparison to those in the no DARVO condition, were less willing to punish the perpetrator and more willing to punish the victim.

Since evidence demonstrates that DARVO is a common response that is related to victim self-blame [16, 17] and is capable of shifting observers' perceptions of sexual violence scenarios in ways that favor perpetrators [20], DARVO may perpetuate victim-blaming beliefs. In the context of sexual violence, beliefs that victims are blameworthy for their own victimization are commonly described as rape myths.

## Rape myths

High rates of sexual violence are sustained by a rape-supportive culture [21]. Rape myths, which describe stereotypical and false beliefs about sexual violence that shift blame and responsibility to victims [22], play a significant role in rape culture. Most research has focused on rape myths in the context of male-perpetrated sexual violence against women [23]. The notion that women compel male perpetrators to commit sexual offenses through their actions or appearance is a common rape myth, as are the beliefs that women lie about being assaulted and that men unintentionally commit sexual assault by way of intoxication or uncontrollable sexual desire. Decades of research on rape myths about male perpetrators and female victims have produced stable findings. In a meta-analysis of 37 studies, researchers identified that men report greater rape myth acceptance than women [23]; the meta-analysis also revealed that individuals' rape myth acceptance is generally related to hostility toward women, victim-blaming attitudes, and acceptance of interpersonal violence. In a separate meta-analysis of studies

investigating the connection between men's sexual violence perpetration and rape myth acceptance, eight out of the nine studies examining this association identified a positive relationship between the two variables [24]. Quantitative research on mock jurors' judgements in sexual assault cases reliably finds rape myth acceptance is associated with greater victim blame and fewer guilty judgements for the perpetrator [25]. Rape myths may also impact rape victims' perceptions of their own experiences of assault. Studies with women who have been sexually assaulted reveal that victims with greater rape myth acceptance are less likely to report their rape [26] and experience greater self-blame [27].

Rape myths create a toxic social environment that is harmful for victims and advantageous for perpetrators. As such, programs seeking to reduce and prevent sexual violence often target rape myths to reduce individuals' rape myth acceptance [28]. Research suggests that programs targeting rape myths can be effective in attenuating individuals' rape myth acceptance [29, 30] although these reductions have been detected primarily in the short period of time following program interventions [31, 32].

DARVO responses in cases of sexual violence strongly echo rape myths that blame victims and explain away the actions of perpetrators. For instance, the belief that victims lie about being assaulted easily maps onto DARVO: it denies that an assault took place, attacks the victim by positing that the allegations are fabricated, and casts the perpetrator as the victim of a false accusation. This rape myth–and DARVO response–can be so effective that some victims of sexual violence have faced criminal charges for making so-called false reports [33]. Rape myths, arguably, are propagated when perpetrators' DARVO responses to sexual violence are internalized and accepted on a society-wide level.

Despite the parallels between DARVO responses and rape myths, these are conceptually distinct concepts. DARVO is an interpersonal behavioral response used as a mechanism of self-defense; it primarily functions to deflect accountability. Rape myths, on the other hand, are widely held cultural beliefs that are rooted in patriarchal violence. An individual using DARVO to evade blame for sexual violence may invoke certain rape myths in the process of using the self-defense tactic, but DARVO does not encompass societal–or even personal– beliefs about sexual violence.

## Current study

Both rape myths and DARVO threaten the wellbeing of sexual violence victims and promote victim-blaming narratives. Moreover, DARVO is an interpersonal behavioral tactic that undermines victims' credibility by portraying victims of sexual violence as dishonest, unreliable, or vindictive. When sociocultural environments support narratives that deny victimization and cast blame on victims, perpetrators are less likely to face consequences for their harmful behaviors and are potentially more likely to continue committing abuses. In this way, perpetrators' use of DARVO may help sustain sexual violence by contributing to a victim-blaming culture and by cultivating social conditions that allow for continued perpetration. Exploring DARVO's connection with attitudes known to contribute to sexual violence (i.e., rape myths) as well as to sexual violence perpetration is critical for understanding this tactic's role in rape culture.

Although DARVO was originally proposed to be a sex offender tactic [14], little empirical work has examined the relationship between DARVO and sexual violence. Moreover, research has not yet directly examined people who use DARVO. Previous studies have either presented experimental vignettes containing DARVO [20] or asked victims to report on their exposure to DARVO-like responses during confrontations [16, 17]. The purpose of the current study was to explore DARVO's connection to sexual violence by investigating the general use of this

tactic in conjunction a behavioral measure of engagement in sexual harassment. The pervasiveness of sexual harassment [1] and its less blatantly violent characteristics (compared to, for example, rape or physically aggressive sexual violence) may allow participants to be more willing to acknowledge and report on sexually harassing behaviors.

The current study aimed to quantitatively test associations between DARVO use, rape myths, and sexual harassment perpetration among two different samples: undergraduates and community members recruited online. As previous DARVO research has been primarily conducted with undergraduate participants [16, 17, 20], it was important to examine DARVO within a community sample of adults for generalizability.

The current study therefore tested the following hypotheses:

H1: Individuals' DARVO use will be positively associated with rape myth acceptance in both the undergraduate (H1a) and community (H1b) samples

H2: Individuals' DARVO use will be positively associated with their participation in sexually harassing behaviors in both the undergraduate (H2a) and community (H2b) samples

Since this study is the first to explore these associations, an exploratory multivariate model predicting Deny, Attack, and Reverse responses from rape myth acceptance and sexual harassment perpetration will also be conducted to provide more granular findings. No specific hypotheses were made for the multivariate analysis. Moreover, given gendered differences in RMA [23] and sexual harassment perpetration [13], we expected to observe higher scores among men than women for these variables. Gender differences in DARVO use were explored without predictions.

## Method

### Participants

**Undergraduate sample.** Undergraduate participants were students from a large public university in the American West. A total of 641 students were recruited from the university's human subjects pool, which is largely populated by students taking introductory psychology and linguistics courses. To prevent self-selection, the survey's listing did not provide specific information about the content of the current study's measures. Participants were therefore unaware of the study's content when they signed up to participate in the survey, which minimizes threats to external validity caused by self-selection [34]. Of the students recruited for the study, two did not complete the survey and an additional 37 did not correctly respond to the survey's two attention check questions. Data from these individuals were excluded, producing a final sample of 602 students.

All students in the sample were adults and reported an average of 19.54 years old ($SD$ = 1.93). Most students identified as women ($n$ = 415, 68.9%) or men ($n$ = 153, 25.4%), with smaller numbers of students identifying as non-binary ($n$ = 19, 3.2%), genderqueer ($n$ = 9, 1.5%), agender ($n$ = 3, 0.3%), or transgender ($n$ = 1, 0.2%). An administrative error led to the unintentional omission of sexual orientation demographic information. Over three-quarters of the students in the sample were white or Caucasian ($n$ = 462, 76.7%). Most students described themselves as either "*Not at all*" ($n$ = 332, 55.1%) or "*A little bit*" religious ($n$ = 192, 31.9%) and as politically liberal ($n$ = 251, 41.8%) or moderate ($n$ = 154, 25.6%). See Table 1 for additional demographic information.

**Community sample.** Community participants were recruited on Amazon Mechanical Turk (MTurk), an online crowdsourcing service used by a growing number of researchers for data collection [35]. On the MTurk platform, researchers can post their internet studies as

**Table 1. Demographic characteristics of undergraduate and community samples.**

| Characteristics | Undergraduate sample | | Community sample | |
|---|---|---|---|---|
| | N = 602 | | N = 335 | |
| | *n* | % | *n* | % |
| Gender | | | | |
| Woman | 415 | 68.9 | 139 | 41.5 |
| Man | 153 | 25.4 | 194 | 57.9 |
| Trans, nonbinary, genderqueer, or agender | 32 | 5.2 | 2 | 0.6 |
| Sexual Orientation | N/A* | N/A* | | |
| Heterosexual | | | 298 | 89 |
| Bisexual | | | 16 | 4.8 |
| Gay or lesbian | | | 10 | 3 |
| Pansexual, asexual, aromantic, or queer | | | 9 | 2.7 |
| Racial/Ethnic identity | | | | |
| White or Caucasian | 462 | 76.7 | 277 | 82.7 |
| African American or Black | 25 | 4.2 | 22 | 6.6 |
| East Asian | 46 | 7.6 | 15 | 4.5 |
| Latino, Hispanic, or Chicano | 78 | 13 | 9 | 2.7 |
| Native American or Native Alaskan | 8 | 1.3 | 8 | 2.4 |
| Biracial | 26 | 4.3 | 6 | 1.8 |
| South Asian | 22 | 3.7 | 5 | 1.4 |
| Jewish | 31 | 5.1 | 2 | .6 |
| Pacific Islander | 19 | 3.2 | 0 | 0 |
| Middle Eastern or North African | 8 | 1.3 | 0 | 0 |
| Self-described | 8 | 1.3 | 0 | 0 |
| Political orientation | | | | |
| Very liberal | 148 | 24.7 | 70 | 20.9 |
| Liberal | 251 | 41.8 | 88 | 26.3 |
| Moderate | 154 | 25.6 | 53 | 15.8 |
| Conservative | 21 | 3.5 | 76 | 22.7 |
| Very conservative | 3 | .5 | 45 | 13.4 |
| Self-described | 23 | 3.8 | 3 | 0.9 |
| Religiosity | | | | |
| Not at all religious | 332 | 55.1 | 134 | 40 |
| A little bit religious | 192 | 31.9 | 81 | 24.2 |
| Moderately religious | 63 | 10.5 | 64 | 19.1 |
| Very religious | 13 | 2.2 | 56 | 16.7 |
| | *M* (*SD*) | | *M* (*SD*) | |
| Age | 19.54 (1.93) | | 39.85 (11.23) | |

*Note.*

*Sexual orientation data were not collected for the undergraduate sample due to administrative error

Human Intelligence Tasks (HITs), which individuals (MTurk "workers") sign up for and complete in exchange for payment. Eligibility criteria for the current study selected for individuals who (1) were aged 18 years or older; (2) were currently residing in the United States; (3) had previously completed 50 HITs on MTurk with a HIT approval rating of 95% or higher. A total of 582 participants were recruited on MTurk using these criteria. The data were screened carefully to help ensure the quality of the data, which led to the implementation of additional

criteria. Low-quality data were identified based on the following: 1) participants who spent less than 9 minutes taking the survey (53 participants eliminated); 2) participants who failed to respond to both attention checks correctly (25 participants eliminated), 3) participants with duplicate responses (9 participants eliminated), 4) suspicious responses to write-in prompts asking participants to describe a time they were confronted by someone else, which belonged to another part of the larger survey (160 participants eliminated). Responses were categorized as suspicious if they were incoherent or nonsensical (e.g., "place usa," and "i got never remember"), were copy/pasted text from the survey instructions or ostensibly from an internet search result (e.g., "There something you didn't do, keep quiet, speak with a lawyer, gather evidence, stay away from your abuser, and comply with the court's orders"), or indicated the participant did not comprehend the prompt to describe a time they were confronted (e.g., "My father passed away"). The resulting number of individuals whose data were included for analyses was 335.

On average, participants were 39.85 years old ($SD$ = 11.23). Nearly the entire sample identified as either men (57.9%, $n$ = 194) or women (41.5%, $n$ = 139). Only 1 participant identified as genderqueer and 1 identified as non-binary. The majority of individuals were heterosexual (89%, $n$ = 298), and approximately 83% of the sample identified as white ($n$ = 277). Over half of participants (60%) identified as at least a little bit religious. Politically, 47.2% identified as either liberal or very liberal; the other half of the sample identified their political beliefs as either moderate (15.8%), conservative (22.7%), or very conservative (13.4%). See Table 1 for detailed demographic characteristics of the community sample.

## Materials

**DARVO responses.** Participants' general DARVO use was measured using a modified version of the short-form DARVO Questionnaire (DARVO-SF; [36]). The DARVO-SF is an 18-item scale that evaluates respondents' exposure to DARVO in confrontational contexts. It contains a series of statements representing denial (3 items), attacks (8 items), and reversals of victim and offender (7 items). In the original DARVO-SF, respondents are asked to report on a time of their choosing when they confronted another person who had committed some form of harm or wrongdoing against them. Respondents then rate the similarity of the items' statements to the responses given by the individuals they confronted.

For the current study, modifications were made so that the scale evaluated respondents' own general DARVO use during a time they were confronted by someone else. Respondents were instructed to think of "the worst or most serious thing" they were accused of doing and were then asked report on this experience. The scale did not impose limitations with respect to when the confrontation occurred, with whom the confrontation occurred, or the specific nature of the confrontation–respondents could report on a confrontation that happened at any time and, as long as it was about the worst or most serious thing they were accused of doing, could be about a range of behaviors. In this way, DARVO use was assessed across a variety of potential contexts. Scale instructions were reworded (emphasis added): "Please rate how similar each statement is to what *you* said in response to the confrontation or accusation." The modified version of this scale, called DARVO-USE, is therefore a self-report measure of respondents' use of denials, personal attacks, and reversals of victim and offender during a time they were confronted. Examples of denial items are, "That never happened" and "Whatever you're saying happened isn't my fault." Attack items on the scale include, "You're acting crazy" and, "No one would believe you if you said anything about it." Finally, examples of Reverse Victim and Offender items are, "I'm the real victim here" and, "You should be apologizing to me." For each item, participants are asked to indicate whether their responses during a time they were confronted were *not at all like this* (1), *somewhat like this* (2), *moderately like*

*this* (3), *very much like this* (4), or *almost exactly like this* (5). Items were scored by averaging responses to the DARVO items. In the undergraduate sample, Cronbach's alpha for all items was .842. Cronbach's alpha for the three denial items was low ($\alpha$ = .312), which may reflect the limited number of items in this subscale (i.e., 3). Reliability coefficients for the Attack ($\alpha$ = .777) and Reverse ($\alpha$ = .791) subscales were acceptable. In the community sample, a similar pattern was found: the DARVO-USE scale produced a reliability coefficient of .931 for all 18 items, the Deny subscale resulted in low reliability (Cronbach's alpha = .446), and the Attack (Cronbach's alpha = .909) and Reverse (Cronbach's alpha = .873) subscales resulted in good internal reliability.

In the current study, which uses data originally collected for multiple projects, participants were randomly assigned using the randomizer function in Qualtrics to report their DARVO use in one of two contexts: during a confrontation over a wrongdoing they committed, or during a confrontation over a wrongdoing they believed they did not commit. This contextual element was included during data collection for the purposes of a separate study and is not part of the current study's hypotheses regarding individuals' general DARVO use. However, to ensure that asking participants to report DARVO use in the two different contexts did not significantly impact the study's findings related to rape myth acceptance and sexual harassment perpetration, correlations and multivariate analyses were computed while both controlling and not controlling for this effect. This set of analyses (controlling for and not controlling for DARVO use contexts) did not substantially differ in their findings for either the undergraduate or community samples. Correlations and multivariate analyses are therefore presented without the contextual variable for the sake of simplicity and clarity. Results for the analyses that control for this variable have been made available here.

**Sexual harassment perpetration.** Participants' engagement in sexually harassing behaviors was measured by Stark and colleagues' shortened Sexual Experiences Questionnaire–Department of Defense version (SEQ-DoD-s) [37]. The SEQ-DoD-s was selected due to its relatively limited number of items and for its good psychometric properties [37]. The original 16-item scale measures how often respondents have experienced sexual harassment throughout their life. For the current study, items were reworded to measure lifetime sexual harassment perpetration frequency. For example, "[How often has someone] Made unwanted attempts to establish a romantic sexual relationship with you despite your efforts to discourage it?" was revised to "[How often have you] Made unwanted attempts to establish a romantic or sexual relationship with someone despite their attempts to discourage you?" Items represented four categories of sexually harassing behaviors: sexist hostility (e.g., "Put someone down or acted condescendingly because of his/her gender"), sexual hostility (e.g., "Made offensive remarks about someone's appearance, body, or sexual activities"), unwanted sexual attention ("Touched someone's body in a way that may have made them feel uncomfortable"), and sexual coercion ("Threatened someone with retaliation if they were not being sexually cooperative"). Three additional items created by Rosenthal et al. [38] were also included. These items capture sexually harassing behavior completed online or through electronic messaging (e.g., "Sent to others or posted unwelcome sexual comments, jokes, or pictures by text, email, social media, or other electronic means?"). Responses to the scale's items were made on a 5-point Likert scale ranging from 1 = *never* to 5 = *many times* and were averaged to create an aggregate sexual harassment perpetration score. Cronbach's alpha for all 19 items was .813 in the undergraduate sample and .955 in the community sample.

**Rape myth acceptance.** Rape myth acceptance–the endorsement of cultural myths about rape that blame victims and excuse perpetrators–was evaluated using the Updated Illinois Rape Myth Acceptance Scale [39]. This measure contains 22 items representing four categories of harmful beliefs about male-perpetrated sexual violence against women: it wasn't really rape,

he didn't mean to, she lied, and she asked for it. Each item is a statement containing a myth about rape. Examples include "If a guy is drunk, he might rape someone unintentionally," "When girls are raped, it's often because the way they said "no" was unclear," and "Rape accusations are often used as a way of getting back at guys." Responses to each item are made on a 7-point Likert scale of agreement, ranging from 1 = *strongly disagree* to 7 = *strongly agree* and were averaged together to create an aggregate rape myth acceptance score. Cronbach's alpha was .908 in the undergraduate sample and .966 in the community sample.

## Procedure

Study measures were administered in an online Qualtrics survey. To recruit students for the undergraduate sample, the survey was posted to the university's human subjects pool research participation system. The online posting invited students to participate in an online survey in exchange for course credit but did not reveal any meaningful details about the survey's content, thus eliminating self-selection based on knowledge of the contents. The community sample was recruited by making the survey link available on MTurk as a HIT, which invited eligible individuals (i.e., adults currently residing in the US who had completed at least 50 previous HITs with a HIT approval rating of 95% or greater) to participate in an hour-long research study about people's experiences with confrontations. No information about the study's hypotheses or measures was available to individuals who viewed the study's HIT, thereby reducing self-selection bias. After signing up for the HIT, individuals were able to access the Qualtrics survey link. Prior to participating in the study, the survey warned individuals they should disconnect from any virtual private networks (VPNs) and that use of a VPN would prevent participation in survey. VPNs can be used by MTurk workers to circumvent location requirements; in some cases, VPN use is associated with server farms located outside the US that produce low-quality data [40]. Individuals interested in the study who did not turn off their VPNs following the warning were automatically directed out of the survey using the protocol described in Winter et al. [41]. Participants in both samples were able to take the survey at a time and place of their choosing on their personal electronic devices. Prior to accessing the survey, participants were provided with online informed consent that detailed their rights as research participants.

All participants were presented with the study measures in the same order (i.e., demographic questions, DARVO-USE, rape myth acceptance, and sexual harassment perpetration) within a larger survey containing personality measures unrelated to the current study's hypotheses. The personality measures appeared prior to the DARVO-USE items. The 5% trimmed mean completion time was 32.17 minutes among the undergraduate sample and 33.84 minutes among the community sample. Upon completion of the study, participants were shown a debriefing statement that described the purpose of the study. Undergraduate participants were then granted research participation credit that could partially fulfill a course requirement. Community participants were provided with a $10 payment through MTurk.

Undergraduate recruitment occurred from 17 February 2022 to 9 December 2022, and community recruitment took place from 26 June 2022 to 31 July 2022. All study procedures were approved by the University of Oregon Office of Research Compliance (Institutional Review Board). Informed consent was obtained electronically from all participants prior to study procedures.

## Data analysis plan

Responses were first evaluated for missing data, which were minimal in both samples. In the undergraduate sample, two responses were missing from two items on the DARVO-USE scale

and four items on the RMA measure. One response was missing from four other items on the DARVO-USE scale, nine other items on the RMA measure, and two of the items on the sexual harassment perpetration measure. In the community sample, the DARVO-USE scale was missing a singular response for four items and two responses for two items. The rape myth acceptance scale had a single missing response for five items, and the sexual harassment scale had a single missing response on 5 items and two missing responses on one item. Given the limited number of missing responses across the two samples, an average score was computed for each participant for each study measure.

Variables were then tested for normality. Community sample scores were distributed relatively normally for each of the variables, ranging in skewness from .480 to 1.44 and ranging in kurtosis from -.440 to .519. As these values are within the range of normality [42], no transformations were computed. Undergraduate scores for the continuous main variables–DARVO total score and its Attack and Reverse subscales, rape myth acceptance, and sexual harassment perpetration–produced notable skew (ranging from 1.67 to 2.50) and kurtosis (ranging from 3.52 to 11.62), with sexual harassment perpetration resulting in the highest levels of skewness and kurtosis. Natural log transformations were completed on these variables in the undergraduate sample to produce more normally distributed data for inferential analyses. Skewness (ranging from .75 to 1.41) and kurtosis (ranging from -.024 to 3.24) of the transformed versions of the variables fell within the range of normality suggested by Hair and colleagues [42], i.e., skewness between -2 and +2, and kurtosis between -7 and +7. The Deny subscale of the DARVO-USE scale produced acceptable skewness (.61) and kurtosis (-.31) values and was therefore not transformed for inferential analyses.

After transformations were computed for the undergraduate sample, the following analyses were then conducted for both samples: descriptive statistics (non-transformed variables were used for undergraduate descriptive analyses), independent-samples t-test to compare men and women's means for study variables, bivariate Pearson correlations to evaluate study hypotheses, and an exploratory multivariate regression model predicting individuals' use of Deny, Attack, and Reverse from their rape myth acceptance and sexual harassment perpetration. As the number of participants identifying as a gender other than man or woman was too small (<5% of the sample) for group comparison, only men and women's means were tested for gender differences. In addition to these analyses, a Fisher transformation was conducted to compare the strength of the DARVO correlations between the undergraduate and community samples.

## Results

### Descriptive findings

**Undergraduate sample.**   Averages for the main study variables were relatively low (all below a value of 2 out of 5 for the DARVO and sexual harassment scales and out of 7 for the rape myth acceptance scales). See Table 2 for means and standard deviations. Results from an independent-samples t-test comparing men and women's scores on study variables are also shown in Table 2. This analysis revealed that, while men and women did not differ in their DARVO use, men reported greater rape myth acceptance ($t$(566) = 8.31, $p$ < .001, Cohen's d = .786) and sexual harassment perpetration ($t$(566) = 6.50, $p$ < .001, Cohen's d = .615) than women.

Most participants responded that they previously used at least some form of denial (73.8%) or reversals of victim and offender (62.1%) during a confrontation. Less than half of participants reported employing attacks (42.4%). About one-third of participants (32.1%) reported using all three DARVO elements (Deny, Attack, Reverse) during a confrontation. Among the

**Table 2. Means and standard deviations for study variables, and independent-samples t-test comparing men and women's scores for the undergraduate sample ($N$ = 602).**

| Variable | M(SD) | | | t-test comparing men and women's scores | |
|---|---|---|---|---|---|
| | All participants (N = 602) | Women (N = 415) | Men (N = 153) | t | Cohen's d |
| DARVO total | 1.44 (.48) | 1.44 (.49) | 1.46 (.45) | .590 | .056 |
| Deny subscale | 1.89 (.78) | 1.88 (.78) | 1.94 (.79) | .742 | .070 |
| Attack subscale | 1.26 (.48) | 1.26 (.50) | 1.27 (.46) | .497 | .047 |
| Reverse subscale | 1.44 (.62) | 1.45 (.63) | 1.46 (.58) | .177 | .017 |
| Rape myth acceptance | 1.65 (.67) | 1.54 (.53) | 2.06 (.86) | 8.31*** | .786 |
| Sexual harassment perpetration | 1.21 (.22) | 1.17 (.18) | 1.30 (.29) | 6.50*** | .615 |

***$p < .001$

undergraduates who provided written details of the confrontation they experienced, 10 individuals described being accused of sexual misconduct.

About half of the sample (50.7%) agreed with at least one rape myth at any level of agreement (*agree somewhat*, *agree*, or *strongly agree*). The most commonly agreed-with rape myth ("When guys rape, it is usually because of their strong desire for sex") was endorsed by 30.8% of participants. The least commonly agreed-with rape myths ("A rape probably didn't happen if the girl has no bruises or marks" and "If the accused 'rapist' doesn't have a weapon, you can't really call it a rape") were endorsed by only 1 participant each.

Approximately 80% of the sample ($n$ = 485) reported engaging in at least one sexually harassing behavior at a frequency of "once or twice" or higher. The most reported sexually harassing behavior was telling sexual stories or jokes that may have been offensive to others, which 59% ($n$ = 355) of participants endorsed. The least common sexually harassing behavior in the sample was threatening someone with retaliation if they were not being sexually cooperative; this was endorsed by only 3 individuals.

**Community sample.** As in the undergraduate sample, averages in the community sample were low, with only two variables achieving a mean above a value of 2. Table 3 contains the means and standard deviations, as well as the results from an independent-samples t-test comparing men and women's scores on study variables. Men and women only differed in sexual harassment perpetration, with men ($M$ = 1.61, $SD$ = .62) reporting higher sexual harassment perpetration than women ($M$ = 1.47, $SD$ = .59), $t(331)$ = 2.13, $p$ = .034, Cohen's d = .24.

**Table 3. Means and standard deviations for study variables, and independent-samples t-test comparing men and women's scores for the community sample ($N$ = 335).**

| Variable | M(SD) | | | t-test comparing men and women's scores | |
|---|---|---|---|---|---|
| | All participants (N = 335) | Women (N = 139) | Men (N = 194) | t | Cohen's d |
| DARVO total | 1.83 (.85) | 1.77 (.83) | 1.88 (.87) | 1.14 | .127 |
| Deny subscale | 2.17 (.96) | 2.07 (.98) | 2.25 (.94) | 1.65 | .183 |
| Attack subscale | 1.71 (.93) | 1.65 (.89) | 1.75(.96) | 1.00 | .112 |
| Reverse subscale | 1.83 (.94) | 1.79 (.93) | 1.87 (.94) | .789 | .088 |
| Rape myth acceptance | 2.74 (1.44) | 2.57 (1.46) | 2.86 (1.42) | 1.85 | .205 |
| Sexual harassment perpetration | 1.55 (.61) | 1.47 (.59) | 1.61 (.62) | 2.13* | .237 |

*$p < .05$

The majority of community participants reporting having used denial (76.4%), attacks (58.2%), and reversals of victim and offender (70.4%) during a confrontation. Nearly half of participants (48.1%) indicated they had used all three DARVO elements (Deny, Attack, Reverse) during a confrontation. An analysis of participants' written descriptions of the confrontation revealed that 6 respondents had been confronted about an issue relating to sexual misconduct.

Most participants in the sample (68.3%) expressed agreement with at least one rape myth included in the scale. Among the community participants, the most commonly agreed-with rape myth ("If a girl initiates kissing or hooking up, she should not be surprised if a guy assumes she wants to have sex") was endorsed by 37% of the sample. A minority (12.5%) expressed agreement with the least-endorsed rape myth ("If the accused 'rapist' doesn't have a weapon, you really can't call it a rape").

Approximately 83% of community participants indicated they had engaged in at least one sexually harassing behavior at a frequency of "once or twice" or higher. Telling sexual stories or jokes that may have been offensive to others, was reported by 68% of participants endorsed, making it the most reported sexually harassing behavior in the community sample. Two items on the sexual harassment perpetration measure ("Threatened someone with retaliation if they were not being sexually cooperative" and "Treated someone badly after they refused to have sex") were the least commonly endorsed, with 19.1% of the sample indicating they had participated in those behaviors at least once.

## Correlations

**Undergraduate sample.** Bivariate correlations yielded positive and significant associations between the three DARVO subscales. Individuals who reported using higher levels of denial during confrontations were also more likely to employ personal attacks and attempt to portray themselves as the real victim. In support of hypotheses H1a and H2a, correlations also indicated positive and significant associations between individuals' total DARVO scores and both rape myth acceptance ($r = .135$, $p = .001$) and sexual harassment perpetration ($r = .128$, $p = .002$). The Attack and Reverse subscales resulted in positive correlations with rape myth acceptance ($r = .146$, $p < .001$ and $r = .130$, $p = .001$, respectively) and sexual harassment perpetration ($r = .124$, $p = .002$ and $r = .131$, $p = .001$, respectively). The denial subscale was not correlated with rape myth acceptance or sexual harassment perpetration. Correlations conducted with the untransformed variables produced the same results, with very minor discrepancies in correlation values. The complete results of the bivariate correlations for the undergraduate sample are shown in Table 4.

**Table 4. Bivariate correlations for study variables among all participants in the undergraduate sample ($N = 602$).**

|  | 1. | 2. | 3. | 4. | 5. |
|---|---|---|---|---|---|
| 1. DARVO total | – |  |  |  |  |
| 2. Deny | .667*** | – |  |  |  |
| 3. Attack | .837*** | .390*** | – |  |  |
| 4. Reverse | .862*** | .350*** | .604*** | – |  |
| 5. Rape myth acceptance | .135** | .035 | .146*** | .130** | – |
| 6. Sexual harassment perpetration | .128** | .037 | .124** | .131** | .286*** |

**$p < .01$.

***$p < .001$.

**Table 5. Bivariate correlations for study variables among all participants in the community sample (*N* = 335).**

|  | 1. | 2. | 3. | 4. | 5. |
|---|---|---|---|---|---|
| 1. DARVO total | – |  |  |  |  |
| 2. Deny | .721*** | – |  |  |  |
| 3. Attack | .947*** | .570*** | – |  |  |
| 4. Reverse | .941*** | .595*** | .822*** | – |  |
| 5. Rape myth acceptance | .597*** | .383*** | .583*** | .563*** | – |
| 6. Sexual harassment perpetration | .650*** | .454*** | .640*** | .591*** | .658*** |

***p < .001.

**Community sample.** Bivariate correlations yielded positive and significant associations among all variables, including each of the three DARVO subscales. As with the undergraduate sample, community participants who reported using higher levels of denial during confrontations were also more likely to employ personal attacks and attempt to portray themselves as the real victim. Correlations between individuals' total DARVO scores and both rape myth acceptance ($r$ = .597, $p < .001$) and sexual harassment perpetration ($r$ = .650, $p < .001$) were also significant, which supports hypotheses H1b and H2b. The Deny, Attack and Reverse subscales resulted in positive correlations with both rape myth acceptance and sexual harassment perpetration. The current sample of community participants produced stronger correlations between DARVO and rape myth acceptance ($z$ = 8.08, $p < .05$) and DARVO and sexual harassment perpetration ($z$ = 9.45, $p < .05$) than did the undergraduate sample. Table 5 shows results for all correlations for the community sample.

## Multivariate analysis

**Undergraduate sample.** A multivariate multiple regression model using General Linear Model (GLM) procedures with ordinary least squares (OLS) estimation was run to evaluate rape myth acceptance and sexual harassment perpetration as statistical predictors of the individual Deny, Attack, and Reverse subscales among the undergraduate participants. The omnibus test for the model indicated rape myth acceptance and sexual harassment perpetration significantly predicted the three DARVO subscales, $F(6, 1194)$ = 3.69, $p$ = .001, Wilk's lambda = .964. Tests of between-subjects effects revealed rape myth acceptance and sexual harassment perpetration were unique and significant predictors of the Attack ($B$ = .10, $p$ = .004 and $B$ = .161, $p$ = .033, respectively) and Reverse ($B$ = .099, $p$ = .017 and $B$ = .218, $p$ = .015, respectively) subscales. Neither rape myth acceptance nor sexual harassment perpetration significantly predicted scores on the Deny subscale. See Table 6 below for the regression coefficients resulting from the multivariate multiple analysis.

**Community sample.** Using data from the community sample, the same multivariate multiple regression model was used to estimate rape myth acceptance and sexual harassment perpetration as predictors of the individual Deny, Attack, and Reverse subscales. The omnibus test for the model indicated rape myth acceptance and sexual harassment perpetration significantly predicted the three DARVO subscales, $F(6, 660)$ = 42.94, $p < .001$, Wilk's lambda = .517. Tests of between-subjects effects indicated rape myth acceptance and sexual harassment perpetration were significant predictors of the Deny, Attack, and Reverse subscales. Table 7 contains the results for this multivariate test in the community sample.

## Discussion

The purpose of the current study was to explore associations between general DARVO use in diverse confrontational contexts, rape myth acceptance, and sexual harassment perpetration in

**Table 6. Regression coefficients for a multivariate multiple regression analysis predicting deny, attack, and reverse subscales from rape myth acceptance and sexual harassment perpetration in the undergraduate sample (*N* = 602).**

| | Deny | | | | |
|---|---|---|---|---|---|
| **Predictor variables** | ***B*** | ***SE*** | ***t*** | ***p*** | **95% CI** |
| Rape myth acceptance | .059 | .095 | .620 | .536 | [-.127, .245] |
| Sexual harassment perpetration | .144 | .206 | .701 | .484 | [-.260, .548] |
| | Attack | | | | |
| | *B* | *SE* | *t* | *p* | 95% CI |
| Rape myth acceptance | .100 | .035 | 2.87 | .004 | [.031, .168] |
| Sexual harassment perpetration | .161 | .075 | 2.14 | .033 | [.013, .168] |
| | Reverse | | | | |
| | *B* | *SE* | *t* | *p* | 95% CI |
| Rape myth acceptance | .099 | .041 | 2.39 | .017 | [.017, .180] |
| Sexual harassment perpetration | .218 | .090 | 2.43 | .015 | [.042, .394] |

undergraduate and community samples. In support of hypotheses, both undergraduates and community adults who use DARVO as a response to being confronted over a range of wrong-doings reported greater acceptance of rape myths (H1a and H1b) and were more likely to perpetrate sexual harassment (H2a and H2b). Otherwise stated, individuals who issue denials, personal attacks, and reversals of victim and offender roles as a means of responding to confrontations are more likely to hold victim-blaming beliefs and sexually harass others. These findings suggest that general DARVO use indicates greater support for and participation in rape culture.

There may be several explanations for the association between individuals' DARVO use during confrontations and rape myth acceptance. The first is that people who use DARVO themselves may be overall more accepting of victim blaming in general. Victim blaming is a prominent feature of both DARVO and rape myths. In DARVO, victim blaming occurs interpersonally and on the individual level through victim-blaming dialogue. On a cultural level, false beliefs about sexual violence that attribute blame to victims are propagated through rape myths. Individuals who are willing to engage in victim blaming during confrontations may also be more likely to accept victim-blaming attitudes, like rape myths. The reverse may also

**Table 7. Regression coefficients for a multivariate multiple regression analysis predicting deny, attack, and reverse subscales from rape myth acceptance and sexual harassment perpetration in the community sample (*N* = 335).**

| | Deny | | | | |
|---|---|---|---|---|---|
| **Predictor variables** | ***B*** | ***SE*** | ***t*** | ***p*** | **95% CI** |
| Rape myth acceptance | .099 | .043 | 2.32 | .021 | [.015, .184] |
| Sexual harassment perpetration | .562 | .102 | 5.53 | < .001 | [362, .761] |
| | Attack | | | | |
| | *B* | *SE* | *t* | *p* | 95% CI |
| Rape myth acceptance | .185 | .035 | 5.34 | < .001 | [.117, .253] |
| Sexual harassment perpetration | .690 | .082 | 8.39 | < .001 | [.528, .852] |
| | Reverse | | | | |
| | *B* | *SE* | *t* | *p* | 95% CI |
| Rape myth acceptance | .200 | .037 | 5.45 | < .001 | [.128, .272] |
| Sexual harassment perpetration | .599 | .087 | 6.91 | < .001 | [.428, .769] |

be true: those who are accepting of victim-blaming attitudes may be more likely to respond with victim blaming in confrontational contexts. Victim blaming itself may be underpinned by a worldview that fails to condemn interpersonal violence. When violent behaviors are not regarded as capable of causing harm, then people who claim to have been harmed by such behaviors are consequently not believable. Supporting this logic, research consistently finds that individuals' acceptance of interpersonal violence is associated with their rape myth acceptance [23].

Acceptance of interpersonal violence may also help explain the relationship between DARVO use and sexual harassment perpetration. Prior research has consistently identified acceptance of interpersonal violence as a predictor of men's sexual aggression perpetration [43]. Although research has yet to explore links between individuals' DARVO use and their acceptance of interpersonal violence, it is plausible that people who do not perceive their violent or otherwise antisocial behavior (such as sexual harassment) as unacceptable are more likely to respond defensively with DARVO. Perceiving their behavior to reflect acceptable conduct, individuals who are accepting of interpersonal violence may be genuinely convinced of their innocence. Consequently, these individuals may be more likely to deny they have done anything wrong, attack their victims for making seemingly unjust allegations of harm, and claim to be the real victim of the situation. It is also possible that DARVO directly benefits people who participate in sexually harassing behaviors, which could result in sexual harassers using DARVO more often than those who do not sexually harass. Because DARVO is an effective tactic for perpetrators [14, 20], people who are inclined to sexually harass may have learned that responding with DARVO allows them to deflect at least some blame and responsibility. This reflects other research that describes how perpetrators of interpersonal violence use denials, engage in victim-blaming, and assume a victimized role when recounting their actions [44–46]. Sexual harassers may therefore adopt DARVO responses to excuse their behaviors, including, potentially, a variety of behaviors outside the scope of sexual harassment.

The exploratory multivariate analysis revealed rape myth acceptance and sexual harassment perpetration significantly predicted attack and reverse responses, but not denials, among the undergraduate participants. The same analyses using the community sample indicated rape myth acceptance and participation in sexually harassing behaviors significantly predicted their use of all three elements of DARVO: denials, personal attacks, and reversals of victim and offender during confrontations. While the current study is not able to offer an empirical explanation as to why deny responses were associated with sexual harassment and rape myth acceptance within the community sample but not the undergraduate sample, it is possible that denial is more weakly associated with problematic behaviors or attitudes in general. Of the three elements of DARVO, denial is the least aggressive response as it does not malign victims to the extent that both attacks or reversals of victim and offender do. As such, denials may be more likely to be used by individuals who do not wish to engage in adversarial relations and those who are generally averse to potentially causing harm. This may partially explain why we did not identify associations between denial and sexual harassment and rape myth acceptance in the undergraduate sample; importantly, though, this association emerged in the community sample. Further research is needed to explore the role of denial to fully understand this particular finding. Overall, however, analyses suggest that people who are more likely to believe rape myths and sexually harass are also more likely to respond to confrontations by attacking the credibility of the person confronting them and by assuming the role of a victim. Both rape myths and sexual harassment are part of the fabric of rape culture. As such, evidence from the current study indicates there is a connection between DARVO and some facets of rape culture.

Comparing the results between the two samples, we found that the community sample resulted in stronger correlations between DARVO and both rape myth acceptance and sexual

harassment perpetration. This difference, while notable, is not central to our hypotheses; however, it suggests differences in some demographic variables may attenuate the strength of the associations between DARVO, rape myth acceptance, and sexual harassment perpetration. Additional DARVO research that focuses specifically on the role of education level, socioeconomic status, and other demographic variables would be better equipped to explore this issue.

As the findings from the current study suggest a connection between DARVO and rape culture, anti-rape advocates may benefit from learning about DARVO to identify and interrupt this tactic when they encounter it. Moreover, education seeking to reduce rape myths and sexual harassment could integrate learning modules about how defensive, victim-blaming tactics like DARVO can potentially contribute to rape culture. Such learning modules could therefore discourage the use of DARVO and DARVO-like responses and instead instruct on more pro-social and less damaging responses.

## Limitations and future directions

Although the current study has several important strengths, including a substantial sample size, minimization of self-selection bias, and samples of both undergraduates and community adults, it has limitations as well. The current study signifies an important first step in exploring DARVO, rape myth acceptance, and sexual harassment, but its findings are based on correlational data. As such, we cannot determine the direction of the relationships identified by analyses–do rape myth acceptance and sexual harassment lead people to use DARVO? Or do defensive response styles characteristic of DARVO precede these variables? The multivariate analysis in the current study statistically predicts DARVO from rape myth acceptance and sexual harassment perpetration, but, in practice, this reveals nothing about the directionality of this association. Longitudinal research would be able to address the question of directionality and offer greater insight into the relationship between DARVO, rape myth acceptance, and sexual harassment.

Related to this issue, the current study did not measure variables that might drive the association between all three variables. As discussed above, variables like a general acceptance of interpersonal violence may underlie people's use of DARVO, endorsement of rape myths, and perpetration of sexual harassment. Other unexamined factors, such as antisocial personality traits or certain developmental contexts, may similarly lead people to engage in the three phenomena examined in the current study. Studies that continue this line of research should include constructs that might provide a clearer theoretical perspective on the relationships identified in the current study.

The current study is the first to evaluate individuals' self-reported DARVO use. However, as with most self-report data on past experiences, respondents' capacity to accurately recall and report the event may vary–some respondents may have a clear and accurate recall of the confrontation they were asked to report on, while other respondents' recollections (and therefore reporting) may be less precise. This issue could be potentially addressed by asking respondents to recall a confrontational experience that occurred within a recent timeframe, such as within that past 6 months. Further, the measure prompts respondents to report on a confrontation about the worst or most serious thing they were accused of doing, but it does not limit what the "worst or most serious thing" should be. Respondents were therefore able to report on a range of experiences, including some experiences which might have been relatively trivial. This is a potential limitation of the current study. Given the current study identified a connection between DARVO use and both rape myth acceptance and sexual harassment perpetration, future research would benefit from exploring DARVO use specifically among people who have been confronted about sexual misconduct.

Participant demographics were reasonably varied with respect to age, political identity, and religiosity, but the sample lacked racial diversity as most participants identified as white. Additionally, the number of nonbinary and trans individuals was relatively small in both samples. Future research that probes DARVO and its connection to rape culture would benefit from sampling larger proportions of non-white and gender diverse individuals to better represent the racial and gendered makeup of people living in the US.

Finally, Cronbach's alpha for the Deny subscale was low for both samples. This may be attributable to a few factors. The first factor may be statistical: The Deny subscale only contains three items and Cronbach's alpha tends to be smaller when computed with fewer items. The second possible factor may be related to the conceptual nature of denial; it is possible that the current Deny subscale captures qualitatively different types of denial. The three items representing denial in the subscale are "Whatever you're saying happened isn't my fault," "That never happened," and "You're remembering it incorrectly." The first item does not necessarily deny an event happened, but it does deny responsibility. The second item flatly denies that an event occurred altogether. The third item, like the first, does not deny something occurred; however, it denies the target's memory of the event in a way that is potentially more antagonistic than the other two items. It is therefore possible that these three items represent distinct facets of denial. Future research on types of denial used in DARVO would offer clarity on this issue.

## Conclusions

DARVO is a defensive perpetrator response that promotes a toxic social environment for victims of sexual violence. Its association with rape myth acceptance and sexual harassment perpetration further suggests it plays a role in promoting or reinforcing a culture that supports sexual violence. Identifying and interrupting DARVO responses may offer anti-sexual-violence efforts a novel and potentially effective means of addressing rape culture.

## Author Contributions

**Conceptualization:** Sarah J. Harsey, Jennifer J. Freyd.

**Data curation:** Sarah J. Harsey, Alexis A. Adams-Clark.

**Formal analysis:** Sarah J. Harsey.

**Investigation:** Sarah J. Harsey, Jennifer J. Freyd.

**Methodology:** Sarah J. Harsey, Alexis A. Adams-Clark, Jennifer J. Freyd.

**Project administration:** Sarah J. Harsey, Alexis A. Adams-Clark.

**Supervision:** Jennifer J. Freyd.

**Validation:** Alexis A. Adams-Clark.

**Writing – original draft:** Sarah J. Harsey.

**Writing – review & editing:** Sarah J. Harsey, Alexis A. Adams-Clark, Jennifer J. Freyd.

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
