## [Decision Letter · Decision Letter 0]

9 May 2024

PONE-D-23-33892Associations between defensive victim-blaming responses (DARVO), rape myth acceptance, and sexual harassmentPLOS ONE

Dear Dr. Harsey,

Thank you for submitting your manuscript to PLOS ONE. After careful consideration, we feel that it has merit but does not fully meet PLOS ONE’s publication criteria as it currently stands. Therefore, we invite you to submit a revised version of the manuscript that addresses the points raised during the review process.

Please ensure to provide a point-by-point response to any / all Reviewer feedback (including rebuttals). Please ensure that your revised submission is either 'tracked changes', or is 'cleaned' but has substantial changes highlighted in a clear way. I am sincerely sorry for the delay in getting this feedback and initial decision back to you.

We look forward to receiving your revised manuscript.

Kind regards,

Christopher James Hand, Ph.D., M.Sc., M.A., PgCAP

Academic Editor

PLOS ONE

Journal Requirements:

Reviewers' comments:

Reviewer's Responses to Questions

**Comments to the Author**

1. Is the manuscript technically sound, and do the data support the conclusions?

Reviewer #1: Partly

Reviewer #2: Partly

2. Has the statistical analysis been performed appropriately and rigorously? 

Reviewer #1: Yes

Reviewer #2: No

3. Have the authors made all data underlying the findings in their manuscript fully available?

Reviewer #1: Yes

Reviewer #2: No

4. Is the manuscript presented in an intelligible fashion and written in standard English?

Reviewer #1: Yes

Reviewer #2: Yes

5. Review Comments to the Author

Reviewer #1: This manuscript presents an exploration of DARVO techniques on sexual harassment, sexual perpetration, and rape myths in MTurk and student samples. Results indicated that there were significant associations between DARVO and the outcomes doe both samples. I commend the authors for their work on this manuscript, but there are a few things that need to be addressed before I would consider it for publication:

1. When discussing rape myth, I think it needs to be clearer that the literature you cite seems to be mainly focused on women as victims. There are a different set of rape myths for men, and we don’t know very much about the rape myths for people outside of the binary. I think this distinction could be more clear in this section of the paper. This should also be made clear in the methods as well.

2. Having demographics tables may help with readability and interpretability of these sections. It is a lot of information to keep in mind and it would be easier to have a table to look back at for quick reference.

3. Were there any questions assessing if the DARVO techniques were used in a sexual harassment/victimization scenario? If not, I think it needs to be clear in the abstract (and maybe even title) that you’re looking at DARVO broadly and not necessarily in a sexual harassment/victimization scenario. I was surprised to find it was a general use of these items when I got this this section as it is currently written. I think this needs to be more clear in the methods and discussion sections as well.

4. Why was the DoD version of the SEQ selected? That version was originally designed to capture these behaviours in the military. It seems like an odd choice for use at a civilian university and with a community sample. More information is needed.

5. What were the other measures and where did these fall within the order that they were presented to participants?

6. Your discussion is currently quite short and not a fulsome discussion of the current results. I woud suggest organizing it by hypothesis and providing more detail throughout. I think it need to be very clear that the DARVO measure here is not necessarily reflective of sexual violence. The way it is currently written seems to assume that or brush over the fact that these could be fairly mundane confrontations described by participants. Their behaviours to these confrontations may not necessarily be the same as if they were confronted with abuse, harassment, or sexual violence. This seems like it would probably be the case, but it is an untested assumption. Also, although it was not the main goal of the study, the differences in demographics bears mentioning in the discussion. You have compared two samples and found there to be different patters, that can’t simply be brushed to the side and not discussed. This section could do with some expansion overall.

7. You are also potentially missing out on the voices of trans and non-binary individuals in your sample, which is also an important limitation.

Reviewer #2: PLOS ONE-Article Review 240328

Thank you for the opportunity to review your work. Generally speaking, understanding the factors that contribute to sexual harassment, or any form of sexual violence, is important. This study examines the relationships between several different factors, including DARVO, rape myth acceptance, and sexual harassment perpetration and compared findings across two sample sources. While there are some strengths, I also believe there are areas where the study/manuscript could be improved. Specifically, the conceptual overlap between rape myths and DARVO should be further explored and untangled in the literature review, elements of the methodology need clarification, and the limitations section should be expanded. Below, I detail my recommendations in the general order of the paper’s outline.

Introduction

Please be more specific about the prevalence of sexual violence against male victims. “Less than a third” could be anywhere between 0 and 33%, making it not very meaningful.

The introduction paragraph/section needs a “so what” paragraph or some paragraph connecting the issue of sexual violence to the research questions posed in the current study, as well as why this study is important.

Given the use of sexual harassment as a variable in analyses, it would make more sense to focus discussion of the problem of sexual violence on this specific type of sexual violence. I expect more people have both experienced sexual harassment and perpetrated it, compared to rape or sexual assault. So describing the frequency/prevalence of sexual harassment could make the problem this paper addresses seem more important, pervasive, and relevant.

Literature review

After reading the literature review, I was unsure what the conceptual difference was between DARVO and rape myths. The authors state on page 6 that DARVO “echoes” rape myths and even offers an example of how, so I am confused how the DARVO responses are different from just communicating rape myth beliefs in a specific situation? It might be helpful to include some of the specific DARVO statements that relate to each component. These are listed in the methods section description of this variable, but including examples of both DARVO and rape myths sooner could help clarify. Also, please clarify if DARVO is used in response to any type of perpetration or confrontation, or is specific to sexual violence-related perpetration and confrontations. In reading this section, because the authors focus on DARVO among sexual offenders, it sounded like this behavioral response was specific to sex offenders/sexual violence. If that is the case, then I am not sure there really is much difference between DARVO and rape myth acceptance. DARVO would simply be the statements that transform rape myths—which are essentially neutralizing definitions—into explicit statements. For example, if a person believes the accidental rape myth, or that men rape accidentally when they are too turned on or intoxicated, then they may say DARVO related statements, like, “I didn’t mean to do,” “It was an accident,” “You should not have worn that dress/made out with me if you didn’t want to have sex,” or “You should have said no more clearly.” My point is that DARVO responses to a sexual violence situation seem likely the result of rape myth acceptance. If the concept of DARVO was developed and studied primarily in the area of sexual violence offending, then I am not sure there is sufficient conceptual difference between DARVO and rape myths. So, unpacking this and clarifying these concepts and their differences is essential.

Methods

I am confused by the confrontation component of the survey (. I am generally very confused by how this component was assessed. I understand how the DARVO behaviors/responses were measured. That is not my concern. My issue is with the confrontation they reflected on when responding to the DARVO questions. Were they exposed to a hypothetical confrontation event and asked how they would respond? Or were they asked to recall a confrontation that they experienced? If this second option, were they told to recall any confrontation or only a confrontation related to sexual harassment? If any confrontation, did the authors control for the nature of the confrontation? I expect that DARVO use might vary depending on if the confrontation was about sexual violence, some other crime, or some less concerning conflict, like who left the lights on in an empty house or some such mundane issue. Also, was there a timeframe for the confrontation, such as, within the last week, last month, or last year? Who was doing the confronting? Was it the victim or some third party? Given the confrontation is what precipitated the DARVO responses, clarifying what this was and how it was measured would be helpful.

Later the authors explain that some cases involved individuals who were innocent, and some involved people who were guilty of whatever action they were confronted about. Then it sounded like they included both groups because the diagnostic tests showed no difference between their responses or the relationship between relevant variables. This is a problem. There are very different reasons why a person who is innocent would use DARVO and why a person who is guilty would use DARVO. The innocent person would reasonable deny their perpetration because they actually didn’t do anything wrong. They also would reasonably claim that they were being victimized by the accusations. If not DARVO, then how else might innocent people respond to confrontation/accusations? My point is that, there is no good theoretical reason that DARVO responses to a confrontation about something you did not do would be indicative of rape myth acceptance or sexual harassment perpetration. So even if statistically, the relationships between these variables are the same in all the groups, the reasons for these relationships should be different, and so the groups should be analyzed separate. I would argue the nonsignificant difference between the groups indicates a conceptual problem with the study’s framework. What should differentiate people who use DARVO because they didn’t do something versus those who use DARVO when they did do something? I understand why RMA might predict DARVO use among the guilty, but why would it predict DARVO use among the innocent? Even the guilty may not believe they are guilty if they truly believe rape myths. That is, a person who believes victims deserve rape if they make out with you first may not believe they perpetrated are rape if foreplay was involved.

Given the sample is sufficiently large to look at only the group who responded to the confrontation about something they did do, I recommend only using that group and excluding the other group (the innocents).

The authors say they used “Multiple Linear Regression” to estimate the effects of RMA and sexual harassment perpetration on DARVO. My understanding of MLR is that it is a general term describe a linear regression that uses multiple predictors and a single outcome. But, there are different ways to estimate these models, such as using OLS or HLM. Can the authors clarify the specific type of regression/test they performed?

The models presented in table 3 and 6 are hard to read. Typically, predictors for each outcome/model are grouped together, and then model level statistics like the Wald X2, log likelihood, and/or R2 are included to show model fit. As presented, it looks like the authors ran 6 models, 3 (one for each DARVO component) using RMA as a predictor and 3 (one for each DARVO component) using sexual harassment perpetration as a predictor. If that is the case, then this is not “multiple” linear regression, as there is only one predictor. You might consider reorganizing the table following the example below:

Variable Outcome 1

B SE t P CI

V1

V2

V3

Metric1 - - -

Metric 2 - - -

Outcome 2

V1

V2

V3

Metric1 - - -

Metric 2 - - -

Outcome 3

V1

V2

V3

Metric1 - - -

Metric 2 - - -

Did the authors control for any of the participants’ demographic characteristics or if the innocent/guilty confrontation condition? If not, they should in the MLR models.

In the discussion the authors explain that, because this is a cross-sectional correlational study, the order of the variables does not matter. I disagree. Ultimately, the framework of the study suggests understanding RMA and DARVO is important for identifying risk factors for sexual harassment. So, sexual harassment is the outcome of interest, and the authors are interested in assessing to what extent RMA and DARVO—two attitude related factors—predict sexual harassment—a behavior. For this reason, I believe that DARVO and RMA should be entered into models as predictors of sexual harassment, along with controls (e.g., demographics, confrontation type, sample type, etc). Doing it this way might reveal that RMA predicts sexual harassment perpetration but DARVO does not, or that RMA mediates the relationship between DARVO and sexual harassment, as the theory might suggest.

Study 1/Study 2 separation

It was unclear why the authors separated this into two studies. I understand that the source of data was different, and accessing the two different samples followed different procedures, but all other aspects of the study appeared to be the same. So, why not simply describe one study, include both data collection procedures, and then describe one set of results. You could always do a split sample analysis splitting the sample of students and MTurk respondents so that you still get to replicate findings in two groups. This just does not seem like two studies to me. It is one study, with two different samples.

The large decrease in the MTurk sample size, after removing problematic responses is a problem. Almost half the sample was dropped. Can the authors clarify if the dropped respondents were substantively different from the final sample?

Discussion

The authors suggest that differences in demographics could account for differences in effect sizes/correlations in the Mturk and student samples. This statement could be better supported by including demographics as controls in the multivariate models.

The debate of which came first—the behavior or the attitude—is ongoing. Still, when it comes to rape myths, the research tends to frame the attitude as happening first. People who are high in RMA are more likely to perpetrate sexual violence, more likely to blame victims, less likely to acknowledge or report their own rape. So, the comment on pg 26, line 524-25 is backwards. RMA/rape culture may contribute to DARVO, rather than DARVO somehow increasing rape myth acceptance.

Limitations

If the “confrontation” was one that the respondent recalled, rather than a hypothetical scenario, then this this a limitation that should be noted. Depending on when the even occurred, respondents could forget their response to confrontation. So there needs to be more clarity on what the confrontation component is, and how it was assessed, and if memory may be an issue.

Typos and grammar issues

P. 4, line 82, 85; p 7 line 164; p 9, line 196; p. 12, line 273; p. 25, line 519;

6. PLOS authors have the option to publish the peer review history of their article (what does this mean?). If published, this will include your full peer review and any attached files.

Reviewer #1: No

Reviewer #2: No

---

## [Author Response · Author response to Decision Letter 0]

11 Jul 2024

Please see the Response to Reviewers document attached to this resubmission.

---

## [Decision Letter · Decision Letter 1]

4 Sep 2024

PONE-D-23-33892R1Associations between defensive victim-blaming responses (DARVO), rape myth acceptance, and sexual harassmentPLOS ONE

Dear Dr. Harsey,

Thank you for submitting your manuscript to PLOS ONE. After careful consideration, we feel that it has merit but does not fully meet PLOS ONE’s publication criteria as it currently stands. Therefore, we invite you to submit a revised version of the manuscript that addresses the points raised during the review process.

We look forward to receiving your revised manuscript.

Kind regards,

Christopher James Hand, Ph.D., M.Sc., M.A., PgCAP

Academic Editor

PLOS ONE

Journal Requirements:

Additional Editor Comments:

Thank you for submitting a clear response to the original reviews as well as an updated manuscript.

The original reviewers have responded with further feedback; the decision is to invite you resubmit after making further Minor Revisions that address the reviewers' feedback.

Please note that a Minor Revisions decision does not guarantee eventual publication.

If you have any queries, please do reach out to me.

Reviewers' comments:

Reviewer's Responses to Questions

**Comments to the Author**

1. If the authors have adequately addressed your comments raised in a previous round of review and you feel that this manuscript is now acceptable for publication, you may indicate that here to bypass the “Comments to the Author” section, enter your conflict of interest statement in the “Confidential to Editor” section, and submit your "Accept" recommendation.

Reviewer #2: (No Response)

2. Is the manuscript technically sound, and do the data support the conclusions?

Reviewer #2: Yes

3. Has the statistical analysis been performed appropriately and rigorously? 

Reviewer #2: Yes

4. Have the authors made all data underlying the findings in their manuscript fully available?

Reviewer #2: Yes

5. Is the manuscript presented in an intelligible fashion and written in standard English?

Reviewer #2: Yes

6. Review Comments to the Author

Reviewer #2: PLOS ONE Article Review—Response to revision

I see the authors completed extensive revisions of the manuscript. I believe their hard work has paid off and the study is much improved! Below I have left a few additional comments/ recommendations that I hope could further improve the manuscript.

I recommend a paragraph at the end of the introduction, before the section on RMA that briefly, in a few sentences, states the research questions an how the authors’ study addresses those questions. As of now, there is no statement of what the study asks, addresses, or does (except for the abstract), until the current study section on page 8. As the reader, it is difficult to follow the reviewed literature, without getting a preview of why it matters. A brief paragraph noting this in the introduction would allow the reader to know what to look for and will prepare them to identify the relationships that the authors set up in the lit review.

DARVO is the clear focus of the study, and is frequently referenced first, such as in the methods section talking about measures. Logically, and for symmetry, it might make more sense to put the DARVO section first in the literature review as well, followed by the section on RMA.

On page 6, the authors comment on the experiences of their participants as evidence that DARVO can be used in many different contexts. Please do not use data/findings from the current study in the literature review justifying the study. Perhaps there is other research that reveals this, that the authors can draw on. Otherwise, they might note the types of incidents that DARVO has been documented in, and then speculate that it may also be used in a wider range. It just seems off to use findings from the study to justify or frame the study (that in the narrative has not been done yet).

On page 8, the authors state the purpose of the study (lines184-186): “The purpose of the current study was to evaluate individuals’ own general use of DARVO as a response to confrontations regarding a range of wrongdoing, including possible incidents of sexual violence as well as a variety of other incidents, and their rape myth acceptance.” This makes it sound like they were interested in exploring the variety of confrontations or accusations that the accused may use DARVO in, such as using a content analysis to categorize characteristics of the confrontation (severity of accusation, relationship between accuser and accused) and see if DARVO use varies by these characteristics. The next statement seems to more accurately reflect the actual purpose of the study and what the authors did: “To further explore DARVO’s connection to sexual violence, the current study also sought to investigate the general use of this tactic in conjunction with a behavioral measure of engagement in sexual harassment.” The second statement is reiterated further, with: “The current study aimed to test associations between DARVO use, rape myths, and sexual harassment perpetration among two different samples: undergraduates and community members recruited online.” These later two statements are clearer and more specific than the first, which is misleading. If a clear statement of what the authors did/the purpose (similar to these second two statements) was included in the introduction, the first statement of purpose presented here may make more sense.

While the authors note that people described a wide range of confrontations, given the theoretical and empirical link of DARVO to sexual violence, it would be helpful to know the proportion of respondents who described a confrontation about a sexual violence related incident.

In the methods section, page 15, please clarify if people were randomly assigned to the confrontation conditions, or otherwise clarify how this assignment was determined.

Research studies show that completing an RMA scale before reviewing a vignette and responding about victim blame or rape proclivity can prime respondents to respond more negatively. Can the authors describe the order of measures in the survey? Did respondents review the DARVO section first, followed by RMA, and then the sexual harassment bit? Was order randomized to prevent priming effects?

Tables 2 and 3 show tests comparing DARVO, RMA, and sexual harassment scores for men and women but the authors never noted possible gender differences as one of their research questions or predicted a relationship among their hypotheses. If this was an area of interest at the outset, please make it clear and include the hypothesis. If this was exploratory, please make that clear why this was explored, and not predicted. Given the number of tests conducted, including those that were not hypothesized, the authors may want to consider using a more stringent alpha for the cutoff.

What was the pre-determined alpha value? I presumed, given it was not stated (or did I miss this?), that it was .05. But on page 25, line 502, they note the p-value for the relationship between RMA and Reverse as .071. Perhaps this is a typo, as the same effect in the table is listed with a p = .017? Please clarify and double-check all your numbers.

The stronger relationship between RMA and DARVO among the community sample could be due, in part, to the higher endorsement of RMA among this group. If I am reading the tables correctly, Mturk-worker RMA was almost twice that of the students, and both were below the neutral point. So, in both cases, the relationship with DARVO could be driven primarily by the relatively few individuals at the higher end of the scale, with scores above the neutral point. It may also be related to age. I am not surprised that none of the demographic characteristics were significant in the model after including RMA, as many studies report that RMA mediates the relationship between demographic characteristics like age, gender, and race and the outcomes of interest, typically victim blame or rape proclivity. But, age is a really good predictor of RMA, and older people tend to have higher RMA than younger people. In unpublished work of my own, I have also found that somewhat different demographic characteristics predict RMA/victim blame among students versus MTurk workers, so I can confirm informally that unmeasured demographic characteristics may matter.

Line 614—the authors write that some underlying factor may lead people to engage in the three variables.” Perhaps “phenomena” instead of “variables” is more precise.

Line 638 refers to “Study 2.”

Typos in lines 550, 555, 580, 603, 653.

7. PLOS authors have the option to publish the peer review history of their article (what does this mean?). If published, this will include your full peer review and any attached files.

Reviewer #2: No

---

## [Author Response · Author response to Decision Letter 1]

10 Oct 2024

Please see the attached Response to Reviewers file in the resubmission.

---

## [Editor Report · Decision Letter 2]

29 Oct 2024

Associations between defensive victim-blaming responses (DARVO), rape myth acceptance, and sexual harassment

PONE-D-23-33892R2

Dear Dr. Harsey,

We’re pleased to inform you that your manuscript has been judged scientifically suitable for publication and will be formally accepted for publication once it meets all outstanding technical requirements.

Kind regards,

Christopher James Hand, Ph.D., M.Sc., M.A., PgCAP

Academic Editor

PLOS ONE
---

## [Editor Report · Acceptance letter]

4 Nov 2024

PONE-D-23-33892R2 

PLOS ONE

Dear Dr. Harsey, 

I'm pleased to inform you that your manuscript has been deemed suitable for publication in PLOS ONE. Congratulations! Your manuscript is now being handed over to our production team.

Kind regards, 

on behalf of

Dr. Christopher James Hand 

Academic Editor

PLOS ONE